# Tear Proteomics Study of Dry Eye Disease: Which Eye Do You Adopt as the Representative Eye for the Study?

**DOI:** 10.3390/ijms22010422

**Published:** 2021-01-03

**Authors:** Ming-Tse Kuo, Po-Chiung Fang, Shu-Fang Kuo, Alexander Chen, Yu-Ting Huang

**Affiliations:** 1Department of Ophthalmology, Kaohsiung Chang Gung Memorial Hospital and Chang Gung University College of Medicine, Kaohsiung 83301, Taiwan; fangpc@cgmh.org.tw (P.-C.F.); a1050276@hotmail.com (A.C.); r06100813@cgmh.org.tw (Y.-T.H.); 2Department of Laboratory Medicine, Kaohsiung Chang Gung Memorial Hospital and Chang Gung University College of Medicine, Kaohsiung 83301, Taiwan; ivykuo@cgmh.org.tw; 3Department of Medical Biotechnology and Laboratory Sciences, College of and Laboratory Sciences, College of Medicine, Chang Gung University, Taoyuan 333323, Taiwan

**Keywords:** dry eye disease, homeostasis, biomarkers, proteomics, binocular discrepancy

## Abstract

Most studies about dry eye disease (DED) chose unilateral eye for investigation and drew conclusions based on monocular results, whereas most studies involving tear proteomics were based on the results of pooling tears from a group of DED patients. Patients with DED were consecutively enrolled for binocular clinical tests, tear biochemical markers of DED, and tear proteome. We found that bilateral eyes of DED patients may have similar but different ocular surface performance and tear proteome. Most ocular surface homeostatic markers and tear biomarkers were not significantly different in the bilateral eyes of DED subjects, and most clinical parameters and tear biomarkers were correlated significantly between bilateral eyes. However, discrepant binocular presentation in the markers of ocular surface homeostasis and the associations with tear proteins suggested that one eye’s performance cannot represent that of the other eye or both eyes. Therefore, in studies for elucidating tear film homeostasis of DED, we may lose some important messages hidden in the fellow eye if we collected clinical and proteomic data only from a unilateral eye. For mechanistic studies, it is recommended that researchers collect tear samples from the eye with more severe DED under sensitive criteria for identifying the more severe eye and evaluating the tear biochemical and proteomic markers with binocular concordance drawn in prior binocular studies.

## 1. Introduction

Dry eye disease (DED) is a multifactorial ocular surface disease characterized by a homeostatic loss of the tear film along with ocular distressing symptoms, of which tear film hyperosmolarity and instability, ocular surface inflammation, and neurosensory deviations play etiological roles [1]. It is a disabling disease that has profoundly impacted the daily life of numerous DED patients worldwide [2]. According to the consensus of the Asia Dry Eye Society [3], the core mechanism of DED, tear film instability, will lead to the loss of tear film homeostasis, which may respond to tear ingredients changes. Therefore, many researchers have devoted to elucidating the loss of tear film homeostasis and the change in tear composition of DED.

To meet the need for diagnosis and to clarify the loss of ocular surface homeostasis in DED, a set of qualified clinical evaluation is required [4]. Ocular Surface Disease Index (OSDI), a subjective assessment, is one of the dry eye questionnaires that included items assessing patients’ health-related quality of life and were evaluated adequately for psychometric properties [5]. Measurement of tear film break-up time may provide a better indicator of the tear film stability to prevent evaporative tear fluid losses [6,7,8]. Compared to the Schirmer test, tear meniscus height (TMH) is a reliable, less invasive, higher reproducible, and cost-effective alternative to determine the tear volume [9,10]. Bulbar redness, caused by dilatation of conjunctival vessels and anterior scleral blood vessels, is a response to ocular surface inflammation. [11,12]. The severity of bulbar redness can be objectively quantified via an automated and standardized image grading system. Moreover, meibography and ocular surface staining are two critical assessment for identifying evaporative dry eye [13,14] and ocular surface disease severity of dry eye [15,16], respectively.

However, most of the studies about DED chose unilateral eye for investigation and drew conclusions based on monocular results [6,17], whereas most of the studies involving tear proteomics adopted the results of pooling tears from a group of DED patients [18,19]. In addition, a qualified assessment of DED must include several clinical tests, as mentioned above, and the priority of these tests has been argued [20]. Moreover, we often find that patients with DED have two eyes of different severity in real-world practice. Furthermore, most clinical tests examine one eye first and then assess the other eye. Therefore, we speculate that even for the same test, the result of the former tested eye may differ from that of the latter.

In order to clarify the binocular issue in the research of DED, this study aimed to identify the binocular concordant and discrepant inferences about the association between the loss of ocular surface homeostasis and changes in tear biomarkers in patients with dry eye.

## 2. Results

### 2.1. Binocular Clinical Performance

A total of 23 female participants who met the inclusion and exclusion criteria of DED was enrolled in this study (Table 1). The mean age of these DED subjects was 48.1 ± 11.1 (ranged from 24 to 68 years), and their mean OSDI score was 37.4 ± 30.0 (ranged from 14.6 to 95.8). Compared to the right eye, the left eye had significantly higher mean non-invasive keratographic average break-up time (NIKBUT_av) and assessable time during the test. Most of the clinical parameters were positively and significantly correlated between two eyes except temporal TMH, nasal bulbar redness, nasal limbal redness, non-invasive keratographic first break-up time (NIKBUT_f), and Oxford staining score. This result indicated that in some ocular surface parameters of dry eye patients, one eye’s clinical performance could not represent that of the other eye.

### 2.2. Binocular Tear Biochemical Markers

There was no significant difference in tear biochemical markers between two eyes of the DED subject, including the concentration of matrix metallopeptidase 9 (MMP-9), the concentration of lactoferrin, and the ratio of MMP-9/lactoferrin. (Figure 1a–c). Meanwhile, there were significant positive correlations in the three tear markers between these subjects’ two eyes (Figure 1d–f). Contrary to some clinical parameters of the ocular surface, the result indicated that one eye’s performance could represent that of the other eye in the three markers of DED.

### 2.3. Binocular Tear Proteomic Spectra

Utilizing the liquid chromatography coupled tandem mass spectrometry (LC-MS/MS), we found no significant difference in the number of identified peptides or proteins in the tears of bilateral eyes (Figure 2a). For commonly identified peptides, which were defined by identification rate ≥ 50% for either eye, although the identification rates of some peptides were different between the right and left eyes, none reached a significant difference in statistics (Figure 2b). There were seven peptides with 100% presentation in both eyes of the DED subjects, including lacritin, immunoglobulin alpha-1 chain C region, lactoferrin, lipocalin-1, lysozyme C, polymeric immunoglobulin receptor, and prolactin-inducible protein.

Six always presented peptides were adopted for mass spectral intensity analysis. There were no significant differences in these peptides’ standardized signal intensities between these DED subjects’ right and left eyes (Figure 3). However, among the six always presented molecules (Figure 4), we found that the signal intensities of lactoferrin and lysozyme C showed a highly significant correlation between two eyes of dry eye patients, but immunoglobulin alpha-1 chain C region did not reach a significant correlation between bilateral eyes. Although bilateral tear spectra of lacritin had a significant Spearman correlation, the beta coefficient was very low (0.0231), and the Pearson correlation was not significant (Figure 4e; r = 5.3 × 10^−4^, *p* = 0.9172). In addition, binocular difference and correlation of polymeric immunoglobulin receptor were shown in the Appendix A.

### 2.4. Association between Clinical Parameters in the Ipsilateral Eye

Comparing the Spearman correlation matrices of bilateral eyes, we found the color code patterns were similar but not the same (Figure 5). The correlation between temporal and central TMHs was significant in the left eye (ρ = 0.48, *p* = 0.0210) but was not significant in the right eye (ρ = 0.14, *p* = 0.5258). Moreover, the correlation between nasal and temporal bulbar redness reached statistical significance in the left eye (ρ = 0.70, *p* = 2.0 × 10^−4^) but did not reached statistical difference in the right eye (ρ = 0.20, *p* = 0.3536). Furthermore, the correlation between meiboscale and Oxford staining score was significantly negative in the right eye (ρ = −0.45, *p* = 0.0328), while this correlation was positive in the left eye but did not reach significance (ρ = 0.12, *p* = 0.6013). Although there were significant positive correlations among NIKBUT_f, NIKBUT_av, and assessable time during the test, the correlation between NIKBUT_f and assessable time was weaker than that between NIKBUT_f and NIKBUT_av, and that between NIKBUT_av and assessable time.

### 2.5. Association between Selected Clinical Parameters and Tear Biomarkers in the Ipsilateral Eye

Comparing the Spearman correlation matrices of bilateral eyes, we also found the color code patterns were very similar but not the same (Figure 6). In addition, the correlations between selected clinical parameters and tear biomarkers were highlighted with the rectangle surrounded by red dash lines at the left lower corner of correlation matrices. For clarity, the Spearman correlation coefficient and the corresponding *p*-value in the rectangle of Figure 6 were shown in Table 2. We found that significant correlations were demonstrated between particular clinical parameters and specific tear biomarkers, and some concordant and discrepant correlations between bilateral eyes were further elucidated (Figure 7).

Concordant binocular associations were shown between MMP-9 and two clinical parameters, central TMH and temporal bulbar redness. MMP-9 concentration had a significant negative correlation with central TMH (Figure 7a), but a significant positive correlation with temporal bulbar redness in bilateral eyes (Figure 7b). In addition, some clinical parameters and tear biomarkers had concordant binocular trends, in which one of the two eyes reached the statistical significance and the other eye had the same positive or negative correlation but did not reach significance in statistics. These connections included a positive correlation between the Oxford staining score and MMP-9 concentration, and negative correlations between NIKBUT_f and MMP-9 concentration, age and lactoferrin concentration, and age lactoferrin signal intensity (Figure 7c–f). However, some clinical parameters and had a discrepant binocular trend, in which one of the two eyes reached the statistical significance but the other eye had an opposite positive or negative correlation. These opposite associations included correlations between age and signal intensity of immunoglobulin alpha-1 chain C region, between temporal bulbar redness and signal intensity of immunoglobulin alpha-1 chain C region, and between Oxford staining score and signal intensity of immunoglobulin alpha-1 chain C region (Figure 7g,h). However, although the Spearman correlation between OSDI and lacritin signal intensity revealed statistically significant, Pearson correlation analysis did not support this connection and showed no statistical significance (Figure 7i).

## 3. Discussion

We hypothesized that bilateral eyes of DED patients might have similar but different tear performance. Thus, we proposed a novel tear proteomics approach to compare binocular ocular surface homeostatic indexes and tear proteomic biomarkers in patients with DED. Moreover, we compared the drawn connections between these clinical parameters and tear biomarkers from bilateral eyes. In this study, we found most clinical parameters and tear biomarkers were not significantly different in DED subjects’ bilateral eyes. In the exception, in terms of clinical performance, NIKBUT_av and assessable time during the NIKBUT test of the left eye were significantly lower than that of the right eye (Table 1). Most ocular surface homeostatic makers and tear biomarkers were significantly correlated between bilateral eyes. The exception included temporal TMH, nasal bulbar redness, nasal limbal redness, NIKBUT_f, Oxford staining score, the signal intensity of immunoglobulin alpha 1 chain C region, and signal intensity of lacritin (Table 1 and Figure 4d,e). The left eye was very similar to the right eye in the association patterns among ocular surface homeostatic markers (Figure 5) and between representative clinical parameters and tear biomarkers (Figure 6). However, discrepant binocular trends drawn in different eyes between clinical parameters and tear biomarkers can be identified, including associations between age, temporal bulbar redness, Oxford staining score, and signal intensity of immunoglobulin alpha-1 chain C region (Figure 7g,h). All of the exceptions and discrepant binocular trends mentioned above suggested that the performance of one eye cannot represent that of the other eye or both eyes.

In our previous study [21], we found a link between ocular surface homeostasis and changes in tear biomarkers based on the performance of the right eye of patients with Sjögren syndrome, including positive correlations between NIKBUT_f and lactoferrin signal intensity, and between bulbar redness and MMP-9 concentration. In this study (Figure 7), we further identified a concordant binocular negative correlation between central TMH and MMP-9 concentration, and a positive correlation between temporal bulbar redness and MMP-9 concentration in patients with DED. However, some significant links are only displayed on the right eye instead of the left eye, including a negative correlation between NIKBUT_f and MMP-9 concentration, a positive correlation between Oxford staining score and MMP-9 concentration, and a negative correlation between age and lactoferrin concentration or signal intensity. In addition, some significant links are only displayed on the left eye, including positive correlations between age, temporal bulbar redness, Oxford staining score, and signal intensity of immunoglobulin alpha-1 chain C region. Therefore, it may lose some important messages if we collected data solely from a unilateral eye.

Koh et al. pointed out forced eye-opening required for the NIKBUT assessment influences the measurement of TMH, possibly due to reflex tear secretion, even in patients with aqueous-deficient dry eye [20]. In our study, compared with those of the right eye, all three NIKBUT parameters of the left eye had about 2 s delay, in which NIKBUT_av and assessable time of NIKBUT test reached statistical significance (Table 1). This result implied the NIKBUT assessment of the first eye could also influence that of the fellow eye. Reflex tear secretion induced by this examination of the right eye may temporarily reinforce the left eye’s tear film stability. However, the continuous binocular test did not show different effects on the measurement of TMH and ocular redness.

In clinical practice, tear film instability (shorter NIKBUT_f), less tear secretion (lower central TMH), inflammatory ocular surface (temporal bulbar redness), and ocular surface staining (Oxford staining score) are commonly used as indicators of DED severity. Although the four severity indices of DED did not show statistical significance between right and left eyes (Table 1), a significant correlation between tear proteomic markers and some of the indicators was various between two eyes (Table 2). In the right eye, we only found a marginal correlation between the signal intensity of lactoferrin and NIKBUT_f. This result implied higher tear lactoferrin could increase the duration of tear film stability. However, in the left eye, there were significant positive correlations between the standardized signal intensity of immunoglobulin alpha-1 chain C region and temporal bulbar redness, and also Oxford staining score. Likewise, the signal of the polymeric immunoglobulin receptor had a positive correlation with temporal bulbar redness. This result suggested that the increase of the tear levels of both immunoglobulin alpha-1 chain C region and polymeric immunoglobulin responded to more severe ocular inflammation. The link was caused by the increased severity of DED with ocular surface erosion.

The tear proteomics approach was limited as a laboratory tool and varied by MS-based proteomic strategies for small volume of tear fluid sample. Patients with DED have a small volume of tears, and many researchers adopted the pooling and eluted tear sample for discovering the subtle changes of tear proteomes. We adopted a standardized procedure via normal saline flush to collect tear samples for obtaining the individual tear mass spectrum of each subject. Moreover, we directly explored the association of lactoferrin-corrected standardized spectral intensities of tear peptides with ocular surface homeostatic markers. For quantifying the mass spectral intensity of a tear protein, some researchers used isotopic labeling protein quantification [22,23], while others used label-free protein quantification [18,24,25,26], which are inherently more complex [18]. Although the low-abundance of tear proteins, such as MMP-9, may not be detectable with this approach, the proteomes composed of abundant tear proteins, such as lactoferrin, can be clarified for each subject via our modality. By avoiding complex functional annotation, our method directly drew the relationship between abundant molecules of tear fluid and ocular surface homeostatic markers.

The flush method is a verified tear fluid sampling procedure [27], but it carried the concerns of diluting the basal tears, stimulating the corneal nerves, and bringing reflex tears to change the tear protein profile. Nevertheless, no approach can guarantee a reflex tears-free sample during tear sampling. Moreover, some patients of DED had nearly no tears for collection. Therefore, we adopted the standardized flush method in this study. No subject reported discomfort during tear sampling, and no apparent reflex tearing of all subjects was found in this procedure. Furthermore, lactoferrin-corrected spectral intensity also minimized the dilution effect of the flush tear sampling.

In a clinical trial of DED treatment, it is wise to treat and collect the data of the eye with more severity since it has a greater room for improvement in dry eye parameters or clinical scores. The researcher may observe more significant progress or faster response to clinical interventions in the eye with more severe DED. However, because DED is a bilateral eye disease, various DED severity indices may indicate a different eye with severe DED. In this study, we found that the clinical performance was associated with biochemical and proteomic markers, in which the associations are similar in bilateral eyes. For mechanistic studies that establish strict criteria for identifying the more severe DED eye, we recommend researchers to collect tear samples from the eye with more severe DED instead of a unilateral eye and evaluate the tear biochemical and proteomic markers with binocular concordance as suggested in this study.

## 4. Materials and Methods

### 4.1. Subjects

This prospective case serial study enrolled female DED patients at the corneal department of Kaohsiung Chang Gung Memorial Hospital (CGMH) from 1 November 2019 to 30 June 2020. Informed consent was obtained from all subjects, and all procedures adhered to the Declaration of Helsinki. Institutional Review Board/Ethics Committee approval (code no. 201900954B0) was obtained from the Committee of Medical Ethics and Human Experiments of CGMH, Taiwan. Subjects were included if they met OSDI > 13, and at least one positive for NIKBUT < 10 s or Oxford staining score > 1. Subjects were excluded if they aged less than 20 years, were in pregnancy, had diabetes mellitus, had acute ocular inflammation or glaucoma, or underwent ocular or eyelid surgery within six months. Both eyes of each subject were evaluated for ocular surface homeostasis and tear biomarkers, including tear biochemical markers and proteome.

### 4.2. Assessment Protocol and Tear Sampling

Each subject completed a dry eye questionnaire, OSDI [28]. Each subject was then assessed for binocular ocular surface homeostasis by a masked examiner, who always checked the right eye first then the left eye. Each patient was examined in the following order: TMH, ocular surface redness scan (R-scan), and NIKBUT, followed by the meibography of the lower eyelid. 

After the above examinations for at least 30 min, tears were always collected from the right eye and then the left eye. Each subject was instructed to lie down on an operation table in a supine position for tear collection. Tear fluid samples were collected according to a standardized eye-flush technique without topical anesthesia [27]. A physician instilled a 60 μL drop of non-preserved normal saline on the cornea with gentle eyelid support by collector and forefinger’s thumb. The subject was instructed to keep the eye open, move the eye around, then look at the nasal side and slightly tilt the head to the lateral about 10–15 degrees. A 20 μL tear sample was obtained under surgical biomicroscope by collecting tears pooling in the fornix near the lateral canthus with an automated pipette (Pipetty 1–20 μL, Icomes Lab Co. Ltd., Iwate, Japan).

The collected tears were immediately centrifugated at 6,000× *g* in a microcentrifuge tube for 10 min at 4 °C and the supernatant fluid was frozen at −20 °C. Finally, the cornea was stained with fluorescence and observed by blue-light illuminating slit-lamp to assess corneal staining via Oxford scheme [29]. The above evaluation procedures were performed in the same order for each participant.

### 4.3. Ocular Surface Homeostatic Markers of DED

Three primary ocular surface homeostatic markers were adopted in this study to determine DED subjects’ binocular clinical performance, including tear secretion, ocular surface inflammation, tear film stability, and meibography. Each patient was noninvasively examined, and all the above homeostatic markers were objectively obtained via a tear film analyzer (Keratograph^®^ 5M, Oculus, GmbH, Wetzlar, Germany).

#### 4.3.1. Quantification of Tear Secretion

With illumination by four infrared diodes of 880 nm wavelength [9], the white ring illumination was deactivated, ensuring a dark background on the Placido ring. The intersected point of the lower eyelid margin and the elongated line connecting the cornea center with 4-o’clock, 6-o’clock, and 8-o’clock limbus was respectively used to determine TMHs of the nasal meniscus, central meniscus, and temporal meniscus. With an integrated rule, the TMHs were obtained. The measurements were repeated three times for 3 s after each blink for both eyes of each subject.

#### 4.3.2. Determination of Ocular Surface Inflammation

The R-scan can detect blood vessels in the conjunctiva and estimates the degree of redness. Each patient underwent the R-scan after shifting the light source to white ring illumination on the Placido ring keratography [11,30]. Each patient was asked to focus on the fixation mark inside the camera to allow the tear film analyzer to capture the entire ocular surface. The built-in software automatically splits the regions as the bulbar nasal region, the bulbar temporal region, the limbal nasal region, and the limbal temporal region. It displays redness level using a grading scale of 0.0–4.0 in 0.1 steps. Six indexes were obtained within 10 s, including nasal bulbar redness, temporal bulbar redness, nasal limbal redness, temporal limbal redness, mean redness, and assessable area.

#### 4.3.3. Assessment of Tear Film Stability

The assessment of NIKBUT was used to determine the tear film stability for each patient. Under 880 nm ring illumination, all patients underwent imaging with the same tear film analyzer [20,31]. The 22 mire rings projected on the cornea were captured by videokeratoscopy. Patients were instructed to concentrate on watching the target, blink naturally, and forcefully retrain their blink for as long as possible.

The video recording started automatically right after the second blink. It lasted for a maximum of 23 s. The NIKBUT was measured automatically and noninvasively as the time between the last blink and the first distortion of Placido rings projected onto the cornea. There are three indexes generated for NIKBUT; the NIKBUT_f records the time at which the first perturbation in the reflected Placido disk pattern occurs, the NIKBUT_av calculates the mean time of all detected perturbations during the test, and the assessable time records the time from the start of the recording to the last blink.

#### 4.3.4. Evaluation of Meibomian Gland Dropout

For standardization, we only examined the meibomian gland in the upper eyelid. Under the illumination 840 nm infrared diode, a meibography was captured by a camera installed in the tear film analyzer’s interferometer. The obtained raw image was transformed into a high contrast image for grading. According to the meiboscale proposed by Pult et al., meibomian gland dropout was classified between grade 0 to grade 4, with the grade increasing one scale every 25% of area loss of meibomian gland [32]. 

### 4.4. Tear Biochemical Markers of DED

Within four weeks after collection, the tear fluid sample was thawed on ice to determine the concentrations of selected tear components and total tear protein and analyze the tear proteome. Lactoferrin and MMP-9, two representative DED biochemical markers, were used to assess the DED-induced change of tear components for these patients. Both lactoferrin and MMP-9 levels in tear fluid were used to diagnose the DED and developed as point-of-care diagnostic tools for DED.

#### 4.4.1. Measurement of the Lactoferrin Concentration of Tear Fluid

The lactoferrin concentrations of all tear samples were measured using a human lactoferrin enzyme-linked immunosorbent assay (ELISA) kit (Catalog no. KA0484, Abnova, Taipei, Taiwan). Following the manufacturer’s instructions, one μL tear fluid sample was diluted with the ELISA buffer. The analysis was performed with a Microplate Reader (Thermo Scientific Multiskan FC with internal ELISA Software; Vantaa, Finland) at a wavelength of 450 nm. The assay was performed twice for each tear sample at room temperature. The concentration of tear components was corrected by a 2.5 × dilution factor estimated by a priori test [21]. Lactoferrin levels in tear fluid are reduced in DED patients. Lactoferrin secreted from the major lacrimal gland binds to iron in tears, exerts anti-microbial, antioxidant, and immunomodulatory activities, and maintains the homeostasis of ocular surface health [33,34].

#### 4.4.2. Measurement of the MMP-9 Concentration of Tear Fluid

The concentrations of MMP-9 of all tear samples were measured by using a human MMP-9 ELISA kit (Catalog no. ARG80129, Arigo, Hsinchu, Taiwan). Under the manufacturer’s instructions, four μL tear sample was diluted with the ELISA buffer to a final volume of 100 μL and carried out at 37 °C. The analysis was performed with the same Microplate Reader (Thermo Scientific Multiskan FC with internal ELISA Software; Vantaa, Finland) at a wavelength of 450 nm. The assay was repeated twice for each tear sample. MMP-9 production increases in response to hyperosmolar situations of the ocular surface, contributing to the disruption of the corneal barrier and increasing the severity of DED [35].

### 4.5. Analysis of Tear Proteome

The analysis of tear proteome was based on our previous work [21]. In brief, the total protein of each tear sample was determined by using the Bradford protein-binding assay (Bio-Rad Protein Assay, Bio-Rad Laboratories Taiwan Ltd., Taipei, Taiwan). For the following analysis of tear proteome, the same 5 μg total protein in each sample was obtained by adjusting the tear fluid sample volume.

#### 4.5.1. In-Solution Digestion of Tear Proteins

Five μL of 0.2 g/L trypsin, ten μL of 100 mM ammonium bicarbonate, and 27.5 μL ddH_2_O were mixed with the sample. At 37 °C, the mixed sample was incubated for six h. Subsequently, 0.5 μL of 0.5 M dithiothreitol solution was added to the sample and incubated for 30 min at 56 °C. By adding 1.5 μL of 0.5 M iodoacetamide, the sample was then incubated in the dark environment for 30 min at room temperature for alkylating reduced cysteine residues.

After that, the sample was added with 0.5 μL of 0.5 M dithiothreitol solution and incubated for 30 min at 37 °C. The sample was then added with 5.0 μL of 0.2 g/L trypsin in a 1:30 mass ratio (trypsin/protein) and reached the last volume of 100 μL with dilution 42.5 μL ddH_2_O. After incubation overnight at 37 °C, the sample was heated for 5 min at 100 °C. Each sample was lyophilized at −80 °C. The sample was restored with 25 μL of 0.1% trifluoroacetic acid. Lastly, three μL of the sample was analyzed via liquid chromatography coupled tandem mass spectrometry.

#### 4.5.2. LC-MS/MS Analysis

According to the Jian study [36], an HCT Ultra ETDII Ion-trap Mass Spectrometer (Bruker Daltonics) interfaced with an UltiMate 3000 nano high-performance liquid chromatography system (Dionex) with a 15 cm by 75 μm C18 column was used. The sample’s peptides were eluted by means of an acetonitrile gradient at a flow rate of 0.3 mL/min.

The mass spectra for the eluted fractions were acquired as successive sets of scan modes. The MS scan gained the intensity of ions in the range of 200 to 2000 m/z, and a specific ion was selected for a tandem MS/MS scan. The centroid MS/MS data of enzyme-digested fragments were collected using HyStar 3.2, Bio-Tools, and WarpLC software (Bruker Daltonics). Then, the data were submitted to a search program (MASCOT) for searching Swiss-Prot databases of Homo Sapiens with the following settings: A mass tolerance of 0.3 Da for precursor and fragment ions, carbamidomethyl cysteine residues as fixed modifications, one missed cleavage acceptable for trypsin digestion, and oxidized methionine residues for an optional improvement.

### 4.6. Determination of Sample Size

A free online calculator developed and maintained by the Clinical and Translational Sciences Institute (CTSI) of UCSF was used to calculate the sample size (https://sample-size.net/sample-size-study-paired-t-test/). According to the preliminary results in binocular NIKBUT_av differences of the first ten patients in this study, we estimated the sample size by adopted the significance level (α) as 0.05, the desired power (1-β) as 0.8, the standard deviation of the change of 4.3, and the estimated effect size of 3. Accordingly, the estimated sample size was at least 16 subjects for identifying the binocular difference, and 23 subjects were determined to be the sample size of this study.

### 4.7. Data Analysis

For comparing the spectral intensity of peptides among different samples, lactoferrin’s spectral intensity was used as reference according to the lactoferrin concentration determined by ELISA. The 1 mg/mL lactoferrin was assigned as 100 equivalent spectral intensity (a.u.), and the equivalent spectral intensity of lactoferrin was calculated for each subject. The standardized signal intensity of a specific peptide was then calculated by the absolute spectral amplitude of the specific peptide divided by the absolute spectral amplitude of lactoferrin from the same mass spectrum and multiplied by the equivalent lactoferrin spectral intensity of the same subject.

The statistical analysis was performed by GraphPad Prism version 8.4.3 for Windows (GraphPad Software, San Diego, CA, USA). The Wilcoxon matched-pairs signed rank test was used to compare the indexes of ocular surface homeostasis, tear biochemical markers of DED, and tear proteome between two DED patients’ eyes. A general linear regression model and the Spearman rank correlation coefficient was used to explore the association among the selected parameters for the same eye and different eyes. A *p*-value of < 0.05 was considered statistically significant.

## 5. Conclusions

Bilateral eyes of DED patients may have similar but different ocular surface performance and tear composition. Discrepant binocular presentation in markers of ocular surface homeostasis and tear proteins suggested that the performance of one eye cannot represent that of the other eye or both eyes. Therefore, in the study for elucidating tear film homeostasis, we may lose some important messages hidden in the fellow eye if we collected clinical and proteomic data only from a unilateral eye.

## Figures and Tables

**Figure 1 ijms-22-00422-f001:**
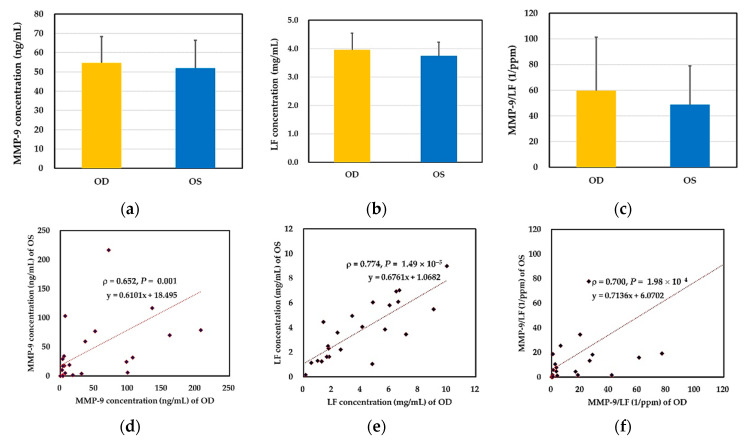
Comparison of the expression of 2 representative biochemical markers in both eyes of dry eye patients (*N* = 23). (**a**) concentration of tear matrix metallopeptidase 9 (MMP-9) (*p* = 0.9881); (**b**) concentration of tear lactoferrin (*p* > 0.9999); and (**c**) concentration ratio of MMP-9 to lactoferrin of each tear sample (*p* = 0.7540). (**d**) correlation of tear MMP-9 concentration; (**e**) correlation of tear lactoferrin concentration; (**f**) correlation of concentration ratio of MMP-9 to lactoferrin of binocular tear samples of each patients. Abbreviation: LF, lactoferrin.

**Figure 2 ijms-22-00422-f002:**
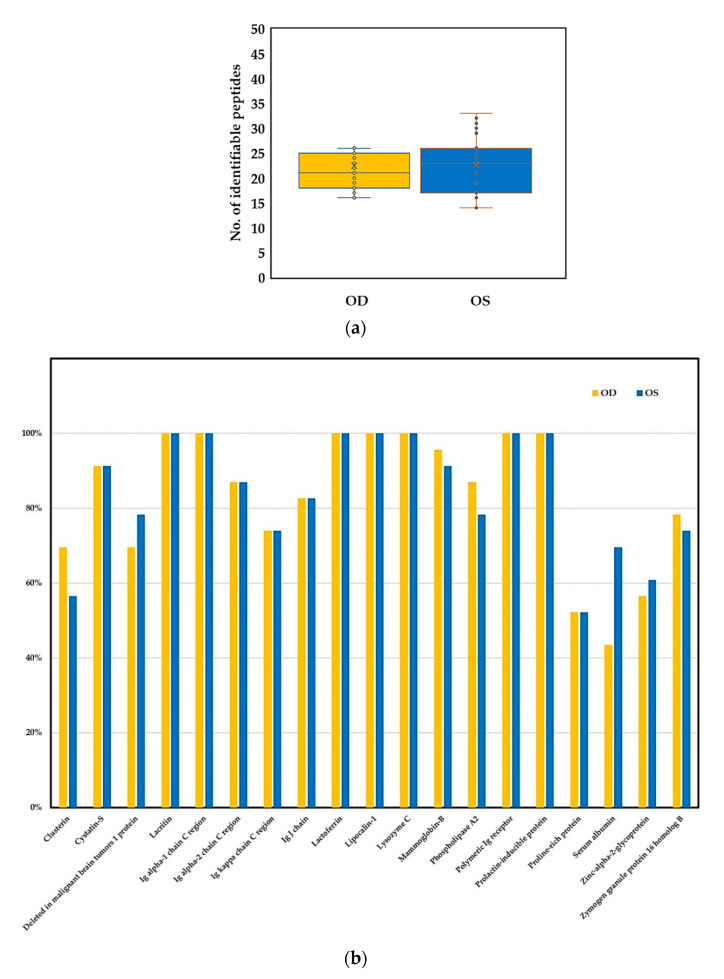
Tear proteome analyzed by the liquid chromatograph coupled tandem mass spectrometry in binocular tear samples of dry eye patients (*N* = 23). (**a**) comparison in the number of identified peptides (*p* = 0.6153); (**b**) comparison in the isolation rate of common identified peptides. Abbreviation: Ig, immunoglobulin.

**Figure 3 ijms-22-00422-f003:**
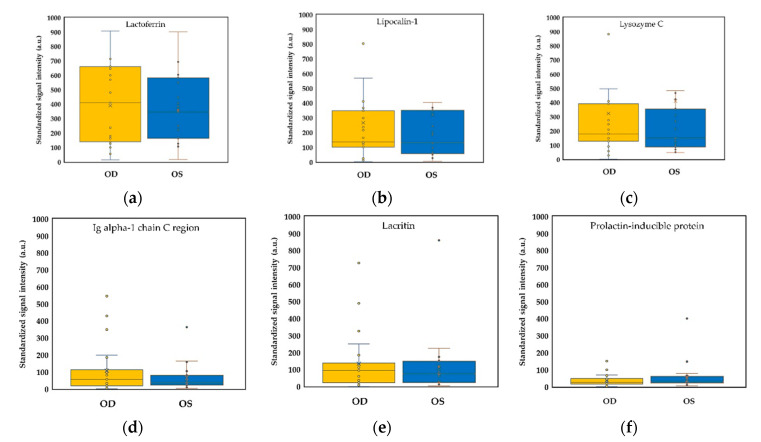
Comparison of lactoferrin-corrected standardized signals for six always presented molecules in binocular tear samples of dry eye subjects (*N* = 23). (**a**) lactoferrin (*p* = 0.8422); (**b**) lipocalin-1 (*p* = 0.6221); (**c**) lysozyme C (*p* = 0.9168); (**d**) Immunoglobulin alpha-1 chain C region (*p* = 0.4820); (**e**) lacritin (*p* = 0.5803); and (**f**) prolactin-inducible protein (*p* = 0.3604). Abbreviation: Ig, immunoglobulin.

**Figure 4 ijms-22-00422-f004:**
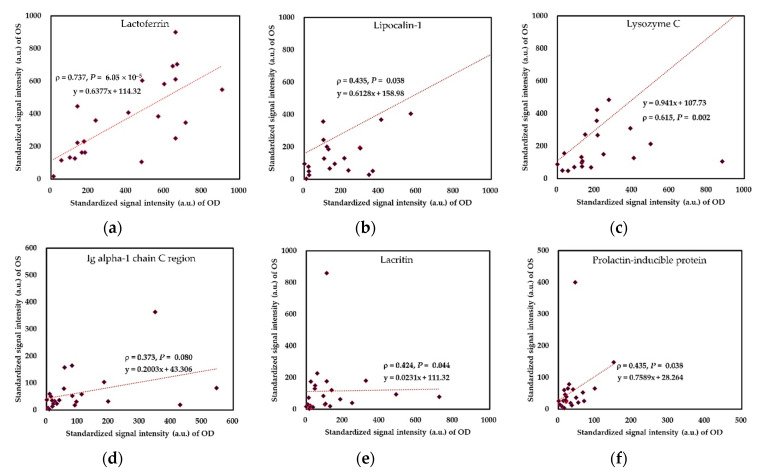
Binocular association of lactoferrin-corrected standardized signals for six always presented molecules in tear samples of dry eye patients. (**a**) lactoferrin; (**b**) lipocalin-1; (**c**) lysozyme C (**d**) immunoglobulin alpha-1 chain C region; (**e**) lacritin; and (**f**) prolactin-inducible protein. Abbreviation: Ig, immunoglobulin.

**Figure 5 ijms-22-00422-f005:**
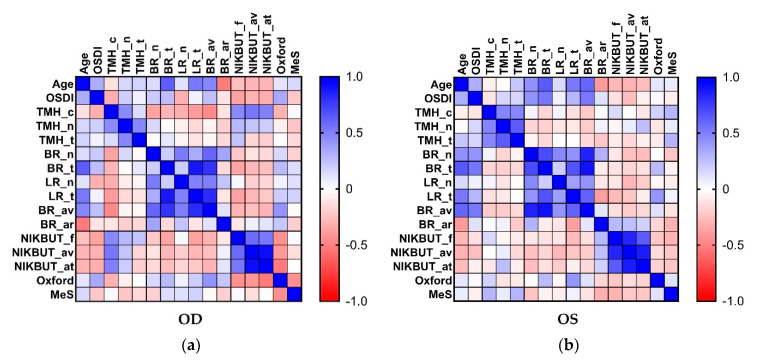
Spearman correlation matrix between different ocular surface homeostatic markers in the ipsilateral eye. (**a**) right eye (*N* = 23); (**b**) left eye (*N* = 23). Abbreviation: OSDI, ocular surface disease index; TMH_c, central tear meniscus height; TMH_n, nasal tear meniscus height; TMH_t, temporal tear meniscus height; BR_n, nasal bulbar redness score; BR_t, temporal bulbar redness score; LR_n, nasal limbal redness score; LR_t, temporal limbal redness score; BR_av, mean redness score; BR_ar, assessable area of ocular redness test; NIKBUT_f, non-invasive keratographic first break-up time; NIKBUT_av, non-invasive keratographic average break-up time; NIKBUT_at, assessable time of non-invasive keratographic break-up test; Oxford, Oxford staining score; and MeS, meiboscale.

**Figure 6 ijms-22-00422-f006:**
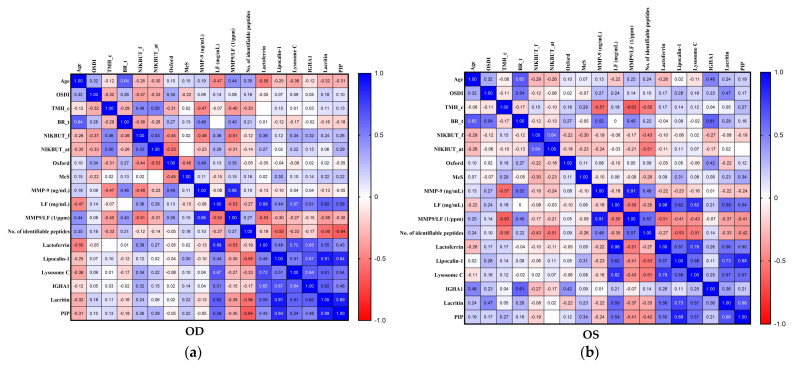
Association between representative ocular surface homeostatic markers and tear biomarkers in the ipsilateral eye. (**a**) right eye (*N* = 23); (**b**) left eye (*N* = 23). Abbreviation: OSDI, ocular surface disease index; TMH_c, central tear meniscus height; BR_t, temporal bulbar redness score; NIKBUT_f, non-invasive keratographic first break-up time; NIKBUT_at, assessable time of non-invasive keratographic break-up test; Oxford, Oxford staining score; MeS, meiboscale; LF, lactoferrin; IGHA1, immunoglobulin alpha-1 chain C region; and PIP, prolactin-inducible protein.

**Figure 7 ijms-22-00422-f007:**
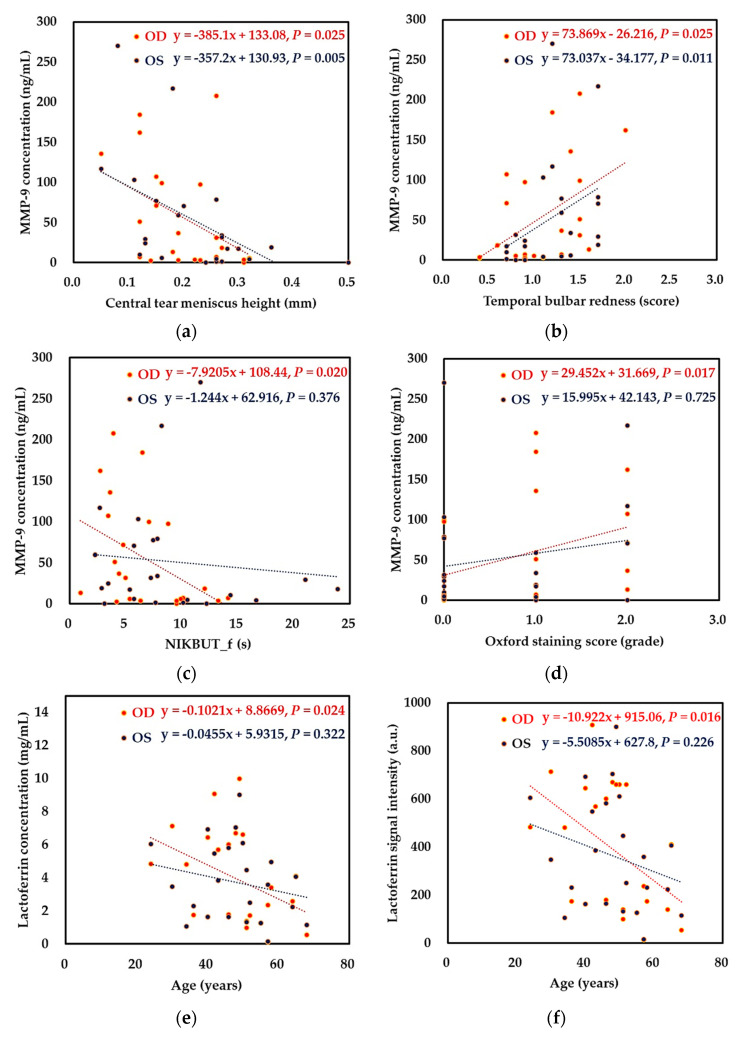
Concordant and discrepant binocular associations between ocular surface homeostatic markers and tear biomarkers in dry eye patients (*N* = 23). (**a**,**b**) concordant binocular associations; (**c**–**f**) concordant binocular trends; and (**g**–**j**) discrepant binocular trends. Abbreviation: NIKBUT_f, non-invasive keratographic first break-up time; IGHA1, immunoglobulin alpha-1 chain C region; and OSDI, ocular surface disease index.

**Table 1 ijms-22-00422-t001:** Binocular clinical performance of dry eye patients.

	OD	OS	*p* Value	
Clinical Parameters ^a^			OD vs. OS ^b^	OD ∝ OS ^c^
Number of eyes	23	23		
Tear meniscus height (mm)				
Nasal meniscus	0.36 ± 0.17	0.40 ± 0.21	0.3244	0.0017
Central meniscus	0.20 ± 0.08	0.22 ± 0.10	0.4349	<0.0001
Temporal meniscus	0.30 ± 0.13	0.33 ± 0.13	0.2253	***0.1048***
Ocular redness score				
Nasal bulbar redness	1.1 ± 0.4	1.1 ± 0.5	0.9401	***0.1542***
Temporal bulbar redness	1.1 ± 0.4	1.2 ± 0.4	0.3272	<0.0001
Nasal limbal redness	0.8 ± 0.4	0.7 ± 0.4	0.7266	***0.1750***
Temporal limbal redness	0.7 ± 0.4	0.9 ± 0.4	0.2100	0.0004
Mean redness	1.1 ± 0.4	1.1 ± 0.4	0.8729	<0.0001
Assessable area	8.0 ± 2.9	8.5 ± 2.6	0.3833	0.0002
Tear break-up time (s)				
NIKBUT_f	6.8 ± 3.5	8.9 ± 5.7	0.3182	***0.1296***
NIKBUT_av	10.6 ± 5.1	12.8 ± 5.7	**0.0233 ***	<0.0001
Assessable time	15.3 ± 7.6	17.5 ± 7.3	**0.0239 ***	<0.0001
Meiboscale	1.3 ± 0.8	1.4 ± 0.8	>0.9999	<0.0001
Oxford staining score	0.8 ± 0.8	0.6 ± 0.8	0.5018	***0.1226***

^a^ All variants were showed in mean ± SD; ^b^ Statistical test by Wilcoxon matched-pairs signed rank test (vs.), *p* < 0.05 was recognized as a significant difference in statistics for comparison between both eyes (bold print and highlighted with *). ^c^ Statistical test by Spearman’s rank correlation (∝), *p* ≥ 0.05 was recognized no statistical correlation or not a good association between both eyes (*p*-value is shown by bold and italic print). Abbreviation: NIKBUT_f, non-invasive keratographic first break-up time; NIKBUT_av, non-invasive keratographic average break-up time.

**Table 2 ijms-22-00422-t002:** Association between ocular surface homeostasis and tear biomarkers in the ipsilateral eye.

	Age	OSDI	TMH_c	BR_t	NIKBUT_f	NIKBUT_at	Oxford	MeS
Right Eye	ρ (*p*-Value)	ρ (*p*-Value)	ρ (*p*-Value)	ρ (*p*-Value)	ρ (*p*-Value)	ρ (*p*-Value)	ρ (*p*-Value)	ρ (*p*-Value)
MMP-9 (ng/mL)	0.19 (0.3790)	0.09 (0.6757)	−0.47 (0.0248)	0.46 (0.0254)	−0.48 (0.0202)	−0.23 (0.2826)	0.49 (0.0167)	0.11 (0.6035)
LF (mg/mL)	−0.47 (0.0239)	0.14 (0.5305)	−0.07 (0.7373)	0.00 (0.9856)	0.36 (0.0893)	0.29 (0.1818)	0.13 (0.5609)	−0.15 (0.5016)
MMP9/LF (1/ppm)	0.44 (0.0367)	0.08 (0.7260)	−0.40 (0.0618)	0.40 (0.0571)	−0.51 (0.0139)	−0.31 (0.1504)	0.35 (0.1058)	0.15 (0.5015)
No. of identifiable peptides	0.35 (0.1002)	0.16 (0.4666)	−0.33 (0.1220)	0.21 (0.3271)	−0.12 (0.5992)	−0.14 (0.5379)	−0.05 (0.8205)	0.16 (0.4734)
Lactoferrin	−0.50 (0.0157)	−0.05 (0.8036)	0.00 (0.9955)	0.01 (0.9613)	0.39 (0.0635)	0.27 (0.2153)	−0.05 (0.8313)	0.02 (0.9312)
Lipocalin-1	−0.25 (0.2475)	0.07 (0.7486)	0.10 (0.6489)	−0.12 (0.5768)	0.12 (0.5960)	0.02 (0.9105)	−0.04 (0.8550)	0.30 (0.1652)
Lysosome C	−0.36 (0.0887)	0.09 (0.6953)	0.01 (0.9588)	−0.17 (0.4367)	0.34 (0.1096)	0.22 (0.3064)	−0.08 (0.7253)	0.10 (0.6611)
IGHA1	−0.12 (0.5802)	0.05 (0.8368)	0.03 (0.8873)	−0.02 (0.9300)	0.32 (0.1317)	0.15 (0.5083)	0.02 (0.9425)	0.14 (0.5190)
Lacritin	−0.32 (0.1429)	0.18 (0.4156)	0.11 (0.6054)	−0.16 (0.4648)	0.24 (0.2642)	0.06 (0.7844)	0.02 (0.9129)	0.22 (0.3153)
PIP	−0.31 (0.1517)	0.10 (0.6366)	0.13 (0.5434)	−0.18 (0.4157)	0.26 (0.2390)	0.29 (0.1819)	−0.05 (0.8080)	0.22 (0.3075)
PMIGR	−0.15 (0.5066)	0.12 (0.5749)	−0.04 (0.8579)	0.07 (0.7589)	0.09 (0.6901)	0.04 (0.8702)	0.02 (0.9176)	0.29 (0.1809)
**Left Eye**								
MMP-9 (ng/mL)	0.13 (0.5481)	0.27 (0.2134)	−0.57 (0.0046)	0.52 (0.0112)	−0.19 (0.3757)	−0.24 (0.2763)	0.08 (0.7255)	−0.10 (0.6646)
LF (mg/mL)	−0.22 (0.3220)	0.24 (0.2756)	0.18 (0.4224)	0.00 (1.0000)	−0.08 (0.7215)	−0.03 (0.8908)	−0.10 (0.6440)	0.06 (0.7809)
MMP9/LF (1/ppm)	0.25 (0.2435)	0.14 (0.5220)	−0.63 (0.0013)	0.45 (0.0304)	−0.17 (0.4506)	−0.21 (0.3362)	0.05 (0.8254)	−0.09 (0.6689)
No. of identifiable peptides	0.24 (0.2793)	0.10 (0.6388)	−0.50 (0.0144)	0.22 (0.3148)	−0.43 (0.0406)	−0.51 (0.0122)	0.09 (0.6666)	−0.26 (0.2293)
Lactoferrin	−0.26 (0.2259)	0.17 (0.4354)	0.17 (0.4424)	−0.04 (0.8733)	−0.10 (0.6608)	−0.11 (0.6123)	−0.05 (0.8244)	0.09 (0.6832)
Lipocalin-1	0.02 (0.9144)	0.28 (0.1922)	0.14 (0.5176)	0.09 (0.6900)	−0.06 (0.7820)	0.11 (0.6308)	0.05 (0.8332)	0.31 (0.1517)
Lysosome C	−0.11 (0.6022)	0.16 (0.4757)	0.12 (0.6001)	−0.02 (0.9297)	0.02 (0.9429)	0.07 (0.7405)	−0.06 (0.7905)	0.08 (0.7039)
IGHA1	0.48 (0.0191)	0.23 (0.2982)	0.04 (0.8558)	0.61 (0.0021)	−0.27 (0.2165)	−0.17 (0.4438)	0.42 (0.0436)	0.09 (0.6699)
Lacritin	0.24 (0.2616)	0.47 (0.0251)	0.05 (0.8181)	0.29 (0.1789)	−0.08 (0.7030)	0.02 (0.9426)	−0.22 (0.3071)	0.23 (0.2847)
PIP	0.19 (0.3748)	0.17 (0.4350)	0.27 (0.2072)	0.16 (0.4794)	−0.19 (0.3940)	0.00 (0.9982)	0.12 (0.5789)	0.34 (0.1151)
PMIGR	0.37 (0.0804)	0.25 (0.2512)	−0.10 (0.6443)	0.51 (0.0127)	−0.30 (0.1609)	−0.20 (0.3625)	0.28 (0.1932)	0.22 (0.3168)

Abbreviation: OSDI, ocular surface disease index; TMH_c, central tear meniscus height; BR_t, temporal bulbar redness score; NIKBUT_f, non-invasive keratographic first break-up time; NIKBUT_at, assessable time of non-invasive keratographic break-up test; Oxford, Oxford staining score; MeS, meiboscale; LF, lactoferrin; IGHA1, immunoglobulin alpha-1 chain C region; PIP, prolactin-inducible protein; PMIGR, polymeric immunoglobulin receptor; and ρ, Spearman correlation coefficient.

## Data Availability

The data presented in this study are available in request from the corresponding author.

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
