# Peer review of "Tear Proteomics Study of Dry Eye Disease: Which Eye Do You Adopt as the Representative Eye for the Study?"

_ijms, 2021, doi:10.3390/ijms22010422_

Round 1
Reviewer 1 Report
Reviewer’s Comments:
The manuscript “Tear Proteomics Study of Dry Eye Disease: Which Eye Do You Adopt as the Representative Eye for the Study?" by Kuo et al demonstrates that bilateral eyes of Dry Eye Disease (DED) patients may have similar but different ocular surface performance and tear proteome.
The following minor edits are recommended:
- Please explain the idea behind using Lactoferrin and MMP-9 as two representative biochemical markers to analyze the DED induced change of tear components in this study.
- Figure 6: Along with the heatmap, please also put the values in a table so it is easier to follow.
- Please mention P-values and sample size (n) in results and in figure legends.
- Please mention the catalog numbers of the chemicals, kits, and instruments used in this study.
- Please check for grammatical errors.
Author Response
Q1. Please explain the idea behind using Lactoferrin and MMP-9 as two representative biochemical markers to analyze the DED induced change of tear components in this study.
A1. We thank the reviewer for this comment. Lactoferrin levels in tear fluid are reduced in DED patients. Lactoferrin secreted from the major lacrimal gland binds to iron in tears, exerts anti-microbial, antioxidant, and immunomodulatory activities, and maintains the homeostasis of ocular surface health. MMP-9 production increases in response to the ocular surface's hyperosmolar situations, contributing to the disruption of the corneal barrier and increasing DED severity. Both Lactoferrin and MMP-9 levels in tear fluid were used to diagnose the DED and developed as point-of-care diagnostic tools for DED. We have explained the idea in Materials and methods (page 13, lines 347-349, 357-360, and 367-369).
Q2. Figure 6: Along with the heatmap, please also put the values in a table so it is easier to follow.
A2. According to the reviewer's comment, we have put the values in a new table (Table 2, pages 7-8).
Q3. Please mention P-values and sample size (n) in results and in figure legends.
A3. We have mentioned P-values and sample size in the results and figure legends (Table 1; page 3, lines 93-94; page 4, lines 106-107; page 5, lines 118-120; page 6, line 139; page 7, line 156; page 10, line 184)
Q4. Please mention the catalog numbers of the chemicals, kits, and instruments used in this study.
A4. We have added the catalog numbers of the chemicals, kits, and instruments used in this study in the section of Materials and Methods (page 14, lines 352, 363, and 373).
Q5. Please check for grammatical errors.
A5. We have checked and corrected the grammatical errors.
Reviewer 2 Report
Review Points>
The authors reported Inconsistency binocular presentation in tear proteomics which is entitled as” Tear Proteomics Study of Dry Eye Disease: Which Eye Do You Adopt as the Representative Eye for the Study”. This paper compared clinical parameters and proteomes between right and left eyes, and compared the association of clinical parameters with proteomes between those eyes. However, data was not appropriately interpreted based on results they got, and I believe the conclusion cannot be drawn by these results. Most biomarkers which are clinically relevant for dry eye in left eye were similar to those in right eye, and association of the clinically relevant biomarkers was similar to each other. Only a few indicators such as IGHA1 and immunoglobulin alpha-1 chain C region showed significant difference, however, relation of those proteins to the dry eye are not known. I don’t see clinically relevant significance in these results. Therefore, it seems not suitable to be published in International Journal of Molecular Science.
Additional Issues>
In Title : the title is too vague to present current research
In Abstract
- The format of the abstract is not acceptable. The methods, results and conclusion were not clearly described.
In Discussions
- The clinical relevance of IGHA1 and immunoglobulin alpha-1 chain C region should be addressed.
- The data should be interpreted accordingly.
Minor Issues>
- There are just some minor type-o’s and grammar errors
Author Response
Q1. This paper compared clinical parameters and proteomes between right and left eyes, and compared the association of clinical parameters with proteomes between those eyes. Most biomarkers which are clinically relevant for dry eye in left eye were similar to those in right eye, and association of the clinically relevant biomarkers was similar to each other. Only a few indicators such as IGHA1 and immunoglobulin alpha-1 chain C region showed significant difference, however, relation of those proteins to the dry eye are not known. I don’t see clinically relevant significance in these results.
A1. The importance of eye selection for intervention or analysis cannot be overemphasized, especially for bilateral eye diseases such as dry eye disease. Therefore, we aimed to explore the possible binocular concordant and discrepant relations between clinical parameters and tear biomarkers in dry eye patients rather than extract the DED biomarkers by comparing them with normal subjects. A concordant result may reinforce the evidence of their links, while a discrepant result often necessitates future studies for further confirmations. We have added a paragraph on multiple binocular linkages between tear biomarkers and clinical parameters in Discussion (pages 10-11, lines 228-241).
Q2. In Title : the title is too vague to present current research
A2. Our title attempts to answer a vital research dilemma in studying binocular dry eye disease using a simple yet concise question.
Q3. In Abstract: The format of the abstract is not acceptable. The methods, results and conclusion were not clearly described.
A3. We have reinforced the abstract's structure in the methods, results, and conclusion (page 1).
Q4. In Discussions: The clinical relevance of IGHA1 and immunoglobulin alpha-1 chain C region should be addressed. The data should be interpreted accordingly.
A4. We have addressed the immunoglobulin alpha-1 chain C region's clinical relevance in Discussion (pages 10-11, lines 235-241).
Q5. There are just some minor type-o’s and grammar errors
A4. We have corrected the type-o's and grammar errors with the help of a native American physician.
Reviewer 3 Report
The manuscript entitled “Tear Proteomics Study of Dry Eye Disease: Which Eye do you Adopt as the Representative Eye for the study?” done by Kuo et al., reports the similarities and differences in clinical and tear biochemical and proteomic profiles between right and left eyes in Dry eye disease patients. This study helps to understand the expression of tear biomarkers in both the eyes of DED patients. However, beyond the potentiality of the manuscript, I have a few minor concerns that the authors may address before going to publish the manuscript.
Minor comments:
- Line 19 in Abstract: Please correct the spelling mistake “makers”.
- Line 36 in Introduction: In line 29, Dry eye disease is already abbreviated as DED. So, throughout the manuscript use DED instead of dry eye disease.
- Line 63: Please write as “mean age”.
- Line 64: please write as “mean OSDI score”.
- Was there any difference in the DED severity between right and left eyes?
- Why only female patients were enrolled in this study? Is there any specific reason for this?
- Please mention based on Binocular Clinical Performance which eye had more severe DED.
- Is the variance in Binocular Clinical Performance due to differences in DED severity between the eyes?
- Line 66, 82, and other places wherever correlation results are mentioned, please mention whether it is a positive correlation or a negative correlation.
- Please discuss Ig lambda-2 chain C region and other 7 identified peptides in relation to DED severity.
- This study would be more meaningful if, the control group with healthy subjects were included.
- Out of 7 peptides with 100% presentation in both eyes of the DED subjects (including lacritin, immunoglobulin alpha-1 chain C region, lactoferrin, lipocalin-1, lysozyme C, polymeric immunoglobulin receptor, prolactin-inducible protein), why polymeric immunoglobulin receptor was not adopted for mass spectral intensity analysis.
- Please mention p-values in the test wherever significant results were mentioned.
- Figure 6 font size is too small to read.
- Was the severity of DED same in both eyes? If not, was it correlated with the studied clinical, biochemical, and proteomic markers?
- There were no significant differences in biochemical markers and proteomic profiles between the two eyes of the DED subject. However, there were some differences in clinical performances. Based on these results, for mechanistic studies can we collect tear samples from one eye or the eye with a more clinical score or severe DED? Please discuss this.
Thanks
Author Response
Q1. Line 19 in Abstract: Please correct the spelling mistake “makers”.
A1. Thanks a lot. We have corrected this word (page 1, line 21).
Q2. Line 36 in Introduction: In line 29, Dry eye disease is already abbreviated as DED. So, throughout the manuscript use DED instead of dry eye disease.
A2. We have abbreviated dry eye disease as DED (page 1, line 42).
Q3. Line 63: Please write as “mean age”.
A3. We have amended this description (page 2, line 69).
Q4. Line 64: please write as “mean OSDI score”.
A4. We have amended this description (page 2, line 70).
Q5. Was there any difference in the DED severity between right and left eyes?
A5. Tear film instability (equivalent to a short NIKBUT-f) is recognized as a critical diagnostic element of DED in both the Tear Film and Ocular Surface Society and the Asia Dry Eye Society. Therefore, we may define the eye with shorter NIKBUT-f as the more severe DED eye. We found that the mean NIKBUT-f of the right eye was 6.8 s was shorter than that of the left eye was 8.9 s (Table 1), but this difference was not statistically significant (P = 0.3182). Therefore, there was no difference in the DED severity between right and left eyes in these subjects.
Q6. Why only female patients were enrolled in this study? Is there any specific reason for this?
A6. Thanks for this question. DED is much more prevalent in women. The primary reason women are at risk for DED is hormonal changes, in which high levels of estrogen and low levels of testosterone contribute to DED. To avoid sex from being a confounding factor, we only included female subjects in this study.
Q7. Please mention based on Binocular Clinical Performance which eye had more severe DED.
A7. As mentioned above (Q5), we think that the shorter NIKBUT-f eye could define the more severe DED eye. Among the 23 patients, we found that 12 patients had shorter NIKBUT-f in the right eye, while 11 patients had shorter NIKBUT-f in the left eye. Therefore, there was also no significant difference in the number of bilateral DED eyes. In addition, there are another three representative indices for DED clinical performance (Table 1), tear secretion [equivalent to the tear meniscus height of central meniscus (TMH_c): 0.20 ± 0.08 vs. 0.22 ± 0.10, P = 0.4349], ocular surface inflammation [equivalent to temporal bulbar redness (BR_t): 1.1 ± 0.4 vs. 1.2 ± 0.4, P = 0.3272], and ocular surface erosion [equivalent to Oxford staining score (0.8 ± 0.8 vs. 0.6 ± 0.8, P = 0.5018)]. We found 9 patients with a shorter TMH_c in the right eye, 9 patients with a shorter TMH_c in the right eye, and 5 patients with an equal TMH_c in both eyes. In addition, there were 7 patients with higher BR_t in the right eye, 11 patients with higher BR_t in the left eye, and 5 patients with equal BR_t in both eyes. Moreover, there were 9 patients with a higher Oxford staining score in the right eye, 5 patients with a higher Oxford staining score in the left eye, and 9 patients with an equal Oxford staining score. Therefore, the three indices also supported no significant differences in the DED severity between right and left eyes.
Q8. Is the variance in Binocular Clinical Performance due to differences in DED severity between the eyes?
A8. The results of NIKBUT-f, central tear meniscus height, temporal bulbar redness, and Oxford staining score revealed no significant difference in the DED severity between right and left eyes, compatible with DED being a bilateral eye disease. However, we found that the NIKBUT-av and the assessment time of tear break-up time of the right eye were significantly shorter than those of the left eye (Table 1). We suggested that the discrepancy may be caused by the influence of forced eye-opening, which is required to assess tear break-up time. We discussed this variance in the third paragraph of Discussion (page 10, lines 220-227).
Q9. Line 66, 82, and other places wherever correlation results are mentioned, please mention whether it is a positive correlation or a negative correlation.
A9. We thank the reviewer for this reminder. We have added “positively correlated” and “a positive correlation” for clarity (page 2, line 72; page 3, line 87).
Q10. Please discuss Ig lambda-2 chain C region and other 7 identified peptides in relation to DED severity.
A10. According to the reviewer’s comment, we recheck the data of the Ig lambda-2 chain C region. We find that we miscalculated the number of positive Ig lambda-2 chain C region (9 positive patients) in the left eye. The number of positive Ig lambda-1 chain C region (7 positive patients) was carelessly added to the positive Ig lambda-2 chain C region. I apologize for this mistake and have revised Figure 2 and the associated statement. Because the Ig lambda-2 chain C region of both eyes was lower than the 50% identification rate (right eye: 30.4% vs. left eye 39.1%), we removed this protein from Figure 2 (page 4).
According to Table 2, we only found a marginal correlation between the signal intensity of lactoferrin and NIKBUT_f in the right eye. This result implied higher tear lactoferrin level could increase the duration of tear film stability. However, in the left eye, there were significant positive correlations between the standardized signal intensity of immunoglobulin alpha-1 chain C region and temporal bulbar redness, and also Oxford staining score. Also, the signal of the polymeric immunoglobulin receptor had a positive correlation with temporal bulbar redness. This result suggested that the increase of the tear levels in both immunoglobulin alpha-1 chain C region and polymeric immunoglobulin responded to more severe ocular inflammation. The link was caused by the increased severity of DED with ocular surface erosion. We have added a paragraph in Discussion (pages 10-11, lines 228-241).
Q11. This study would be more meaningful if, the control group with healthy subjects were included.
A11. Thanks for this comment. This study aimed to identify the binocular concordant and discrepant inferences based on the association between the loss of ocular surface homeostasis and changes in tear biomarkers in patients with DED. Therefore, we did not include healthy subjects who are used to compare DED patients for the identification of novel DED markers. However, we believe that the binocular issue may also exist in healthy subjects.
Q12. Out of 7 peptides with 100% presentation in both eyes of the DED subjects (including lacritin, immunoglobulin alpha-1 chain C region, lactoferrin, lipocalin-1, lysozyme C, polymeric immunoglobulin receptor, prolactin-inducible protein), why polymeric immunoglobulin receptor was not adopted for mass spectral intensity analysis.
A12. We only adopted the 6 peptides because of positive correlations among the 7 tear peptides with 100% presentation in both eyes of the DED. We thought it was enough to demonstrate our binocular concerns in DED by elucidating the associations between 6 representative peptides and clinical parameters. Moreover, under the consideration of a figure layout, it is not aesthetic to show 7 plots in a figure. For more rigor, we compared the binocular standardized signal intensities of polymeric immunoglobulin receptor (29.1 ± 32.7 vs. 23.8 ± 30.7, P = 0.3972) and showed the binocular correlation for this peptide (r = 0.611, P = 0.002) in a supplementary file (Supplementary file 1).
Q13. Please mention p-values in the test wherever significant results were mentioned.
A13. We have added the P-value in results and figure legends wherever they were significant (page 3, lines 91-93; page 4, line 107; page 5, lines 118-120).
Q14. Figure 6 font size is too small to read.
A14. Thanks for this comment. We have amplified the critical correlations revealed in this figure (highlighted with a rectangle surrounded by red dash lines at the left lower corner of correlation matrices) as a new table (Table 2, pages 7-8). Moreover, we have added the standardized signal intensity of polymeric Ig receptor to this table to clarify the association between this tear peptide and binocular clinical performance.
Q15. Was the severity of DED same in both eyes? If not, was it correlated with the studied clinical, biochemical, and proteomic markers?
A15. Thanks for the question. From the replies in Q5 and Q7, we knew the DED severity of the right eye group was the same as that of the left eye group. Therefore, we did not select the severe eye to correlate with their clinical, biochemical, and proteomic markers.
Q16. There were no significant differences in biochemical markers and proteomic profiles between the two eyes of the DED subject. However, there were some differences in clinical performances. Based on these results, for mechanistic studies, can we collect tear samples from one eye or the eye with a more clinical score or severe DED? Please discuss this.
A16. In a clinical trial of DED treatment, it is wise to treat and collect the data of the eye with more severity since it has a greater room for improvement in dry eye parameters or clinical scores. The researcher may observe a greater magnitude of improvement or faster response to clinical interventions in the eye with more severe DED. However, because DED is a bilateral eye disease, various DED severity indices may indicate a different eye with severe DED. In this study, we found that the clinical performance was associated with biochemical and proteomic markers, in which the associations are similar in bilateral eyes. For mechanistic studies that establish strict criteria for identifying the more severe DED eye, we recommend researchers collect tear samples from the eye with more severe DED instead of a unilateral eye and evaluate the tear biochemical and proteomic markers with binocular concordance as suggested in this study. We have added a paragraph in Discussion (page 11, lines 262-271). We thank the reviewer a lot for this comment.